# DRLLA: Deep Reinforcement Learning for Link Adaptation

**Florian Geiser** [1,2,*], **Daniel Wessel** [1,*], **Matthias Hummert** [3], **Andreas Weber** [4], **Dirk Wübben** [3], **Armin Dekorsy** [3] **and Alberto Viseras** [1,*]

1 Motius, 80807 München, Germany
2 Electrical and Computer Engineering, Technical University of Munich, 80333 München, Germany
3 Department of Communications Engineering, University of Bremen, 28359 Bremen, Germany
4 Nokia Bell Labs, 81541 München, Germany
* Correspondence: florian.geiser@tum.de (F.G.); daniel.wessel@motius.de (D.W.); alberto.viseras@motius.de (A.V.)

**Abstract:** Link adaptation (LA) matches transmission parameters to conditions on the radio link, and therefore plays a major role in telecommunications. Improving LA is within the requirements for next-generation mobile telecommunication systems, and by refining link adaptation, a higher channel efficiency can be achieved (i.e., an increased data rate thanks to lower required bandwidth). Furthermore, by replacing traditional LA algorithms, radio transmission systems can better adapt themselves to a dynamic environment. There are several drawbacks to current state-of-the-art approaches, including predefined and static decision boundaries or relying on a single, low-dimensional metric. Nowadays, a broadly used approach to handle a variety of related input variables is a neural network (NN). NNs are able to make use of multiple inputs, and when combined with reinforcement learning (RL), the so-called deep reinforcement learning (DRL) approach emerges. Using DRL, more complex parameter relationships can be considered in order to recommend the modulation and coding scheme (MCS) used in LA. Hence, this work examines the potential of DRL and includes experiments on different channels. The main contribution of this work lies in using DRL algorithms for LA, optimized for throughput based on a subcarrier observation matrix and a packet success rate feedback system. We apply Natural Actor-Critic (NAC) and Proximal Policy Optimization (PPO) algorithms on simulated channels with a subsequent feasibility study on a prerecorded real-world channel. Empirical results produced by experiments on the examined channels hint that Deep Reinforcement Learning for Link Adaptation (DRLLA) offers good performance indicated by a promising data rate on the additive white Gaussian noise (AWGN) channel, the non-line-of-sight (NLOS) channel, and a prerecorded real-world channel. No matter the channel impairment, the agent is able to respond to changing signal-to-interference-plus-noise-ratio (SINR) levels, as exhibited by expected changes in the effective data rate.

**Keywords:** machine learning; mobile communication; reinforcement learning; link adaptation; channel observation

## 1. Introduction

Mobile communication systems have recently started to benefit from different applications of Machine Learning (ML). Link Adaptation (LA) is one candidate for potential improvements, as the currently used approaches have not been enhanced drastically over the last couple of years. LA aims to find suitable modulation and coding schemes (MCSs) for data transmission considering the quality of the radio link. The well-known outer loop link adaptation (OLLA) relies on relatively simple decision-making using a look-up table and acknowledgment or negative acknowledgment (ACK/NACK) feedback [1]. Signal-to-interference-and-noise-ratio (SINR) is used as a robust, but low-dimensional proxy for the channel state [2]. Furthermore, state-of-the-art approaches typically use non-adaptive, precomputed offline link models [3,4]. In comparison, a deployment in the real world could

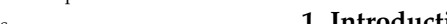

benefit from dynamic and adaptable link models due to the broad spectrum of changing situations offered by some environments.

Reinforcement learning (RL), and especially deep reinforcement learning (DRL), which utilizes neural networks for decision-making, is a promising candidate due to its overall optimization strategy and customizable incentives via reward function. Our proposed solution utilizes DRL to build a next-generation LA approach. DRL allows for handling of multiple inputs while finding links in a high-dimensional solution space.

In this paper, we propose a DRL algorithm called "Deep Reinforcement Learning for Link Adaptation" (DRLLA) for a next-generation mobile telecommunication system that can be deployed on individual base stations. DRLLA benefits from a more dynamic approach considering multiple observable states, such as the environment or the physical characteristics of a wireless transmission system.

A single DRLLA instance was tested on a local machine using a simulation environment to assess the proposed vision of an edge deployment. We implemented DRLLA using OpenAI Gym [5] with an integrated GNU Radio [6] simulation environment. The approach was evaluated on several different channels, including additive white Gaussian noise (AWGN), non-line-of-sight (NLOS), and a prerecorded real-world channel. After analyzing the outcome, we propose the next steps necessary before a deployment.

In this paper, Section 2 provides an overview of recent efforts made in the area of ML for communications considering related or similar approaches. Section 3 provides background knowledge whereas Section 4 presents the proposed DRLLA approach. Section 5 includes the results of our evaluation, which are then discussed in Section 6. Finally, Section 7 concludes this paper and proposes potential improvements.

## 2. Related Work

Improving the performance of LA algorithms is a very relevant problem and has spawned increasing interest in research over the past few years. In a recent paper, Park et al. propose an algorithm that chooses MCSs based on the received channel coefficients and previous packet information [7]. These metrics are translated to SNR values, which are used to derive an MCS with preferably high throughput. They then compare their adaptive preamble-based algorithm to the automatic fallback rate approach as well as the ideal LA scenario. In contrast with Park et al., who aim for a high data rate, Shariatmadri et al. focus on reducing the error rate to improve LA for ultra-reliable communication [8]. To do this, they propose a simple LA scheme that is competitive with optimal LA schemes. In contrast, we focus on achieving a high data rate, accepting the possible negative impact on other transmission characteristics.

Saxena et al. [1] extend Thompson Sampling for the LA use case. Their Latent Thompson Sampling (LTS) method is based on a Bayesian multi-armed bandit approach and observes a low-dimensional latent state model in the form of SINR, while ACK and NACK are used to give feedback on the previous transmission. Mandelli et al. [9] present Training of Outer Loop Link Adaptation (TROLL) to enhance the already simple and robust OLLA implemented in current wireless systems. The approach relies on the baseline OLLA but adds minor modifications, such as gradient detaching to improve the initialization and a channel quality indicator (CQI) correction term (CCT) to correct SINR estimates. Compared to other proposed LA approaches, TROLL is simple to implement and only slightly increases complexity. Our DRLLA approach takes the next step and introduces enhancements and advantages, such as a continuous learning characteristic enabled by RL, which which leads to only a moderate increase in complexity.

Unlike the approaches by Saxena et al. and Mandello et al., the CQI prediction approach using a hidden Markov model (HMM) by Ramezani et al. [10] does not require ACK/NACK feedback. They use an HMM to predict the next CQI through an estimated probability density function (PDF) for the SINR. This approach helps deal with highly dynamic channels, one of the main challenges faced by LA. In contrast, we focus on a combination of SINR and PER for our feedback system during the training procedure.

In [11], Zubow et al. explore using RL for LA, considering the received signal strength indicator (RSSI) as the observed state. A middle layer was created to enable simpler interactions between the signal processing toolkit GNU Radio and their RL models, allowing the Gym framework to interact with a prebuilt flowgraph. In this paper, we build on the open-source GrGym framework, which emerged from the work in [11].

### 3. Background

DRLLA finds itself at the intersection of mobile telecommunication and machine learning. To provide a better understanding of the proposed concept, the following Subsections introduce the objectives of LA, the idea behind RL, and the tools used.

### 3.1. Link Adaptation

LA aims to find the optimal modulation and coding scheme (MCS) for a given channel quality. Usually, each communication standard has its own MCS table, containing different combinations of signal modulations and signal codings. They are sorted in an ascending order based on their theoretical data rate and can therefore be referred to by their MCS index.

In traditional communication systems, the channel quality indicator (CQI) is used as a metric for choosing an appropriate MCS. The user equipment (UE) estimates a CQI value based on the signal-to-interference-plus-noise-ratio (SINR) of the downlink signal [8]. After communicating the CQI to the base station, an MCS can be chosen and used for the following wireless data transmission. To enhance this typical LA approach, we propose using reinforcement learning.

### 3.2. Reinforcement Learning

Reinforcement learning (RL) has recently attracted a lot of attention. It is one of the three main categories of machine learning, alongside supervised and unsupervised learning. The intention is to have an agent learn how to achieve goals in a complex environment subject to uncertainty.

RL consists of two main components: an environment and an agent. The environment encapsulates the problem statement and the rules about the world that can be observed through a state $S$. The agent observes the state and interacts with the environment through a predefined set of actions $A$.

The interaction between the environment and the agent can be broken down into four steps:

1. The agent receives an observable state $S$, provided by the environment.
2. The agent executes an action $A$, based on the received observation.
3. The environment provides a reward $R$, indicating how favorable the performed action was.
4. The agent uses the reward $R$ to tune its action selection process by maximizing the cumulative return.

An action $A$ carried out by the agent might change the environment's state $S$. This process is repeated until the agent's performance converges. The output of the RL training process is an agent that has learned over time by interacting with the environment.

As a further enhancement, we apply an actor-critic (AC) approach that combines both a value-based and a policy-based learning strategy, as displayed in Figure 1. This work examines a natural actor-critic (NAC) algorithm and a proximal-policy optimization (PPO) algorithm, both of which build on the AC approach.

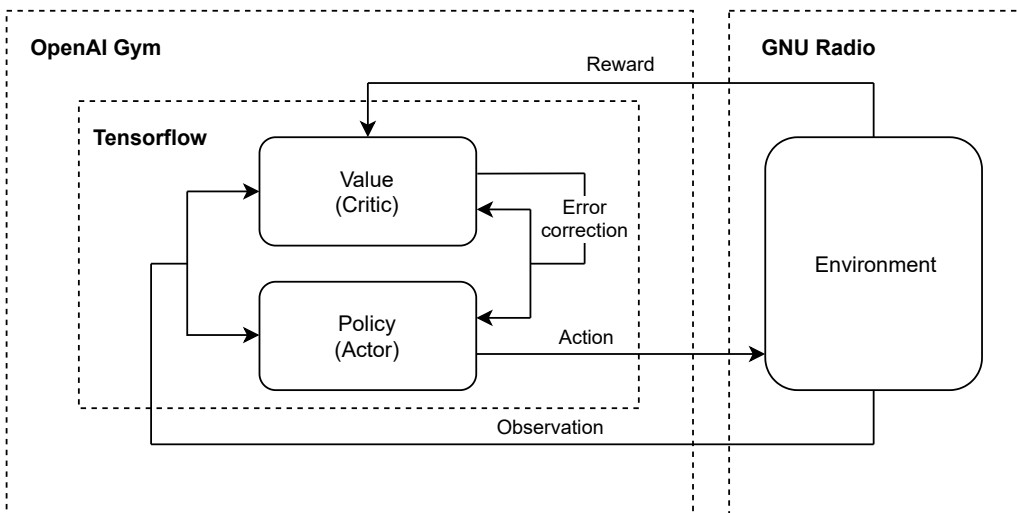

**Figure 1.** Actor-critic reinforcement learning approach with used toolkits.

*3.3. Tools*

The presented algorithms rely on several libraries and toolkits. The following Subsections cover the most important libraries for enabling the training of the agent and the communication with the simulation environment, as shown in Figure 1.

### 3.3.1. GNU Radio

The open-source software development toolkit GNU Radio is used for signal-processing and software-defined radio. Prebuilt and custom blocks (written in Python or C++) can be connected in a flowgraph to define the flow of information and processing in a communication or signal processing pipeline. We define our communication pipeline and connect it using the GrGym middle-layer built in [11] to OpenAI Gym. Since the flowgraph compiles to Python code, Gym can manipulate the variables and run the flowgraph by using the GrGym middle-layer. State $S$ and reward $R$ values are passed from the environment to the RL agent using blocks built for the asynchronous messaging library ZeroMQ. The recommended action $A$ is set as a variable in GNU Radio, thanks to GrGym.

### 3.3.2. OpenAI Gym

Gym [5] is a toolkit originally open-sourced by OpenAI for building and training reinforcement learning applications. It abstracts the environment away from the learning agent by defining a common communication API between the two elements. The vision of Gym is to evolve into an RL framework shared by researchers and scientists to enable easier comparisons of different methods and results. It allows the standardization of approaches and environments.

### 3.3.3. GrGym

A wrapper is required to implement the expected communication API with the agent to facilitate the use of GNU Radio as an RL environment in Gym. GrGym [11] implements this bridge between Gym and GNU Radio. It encapsulates GNU Radio and makes it accessible like any other Gym training environment. This allows the standard usage of the Gym framework for training RL models on a GNU Radio-based environment.

## 4. Method

We propose an online learning approach based on DRL to solve the LA task. This work takes the first step in evaluating the effectiveness of using DRL for LA with a large and less compromised observation space. The goal of high throughput is achieved based on the surveillance of each individual subcarrier state.

As shown in Figure 2, the agent observes the state *S* of the system on the receiver side and proposes an MCS recommendation to the transmitter. The transmitter then uses the MCS while the resulting PER on the receiver side determines the reward *R*. The agent uses the reward *R* to adjust its decision-making process, modeled as a neural network.

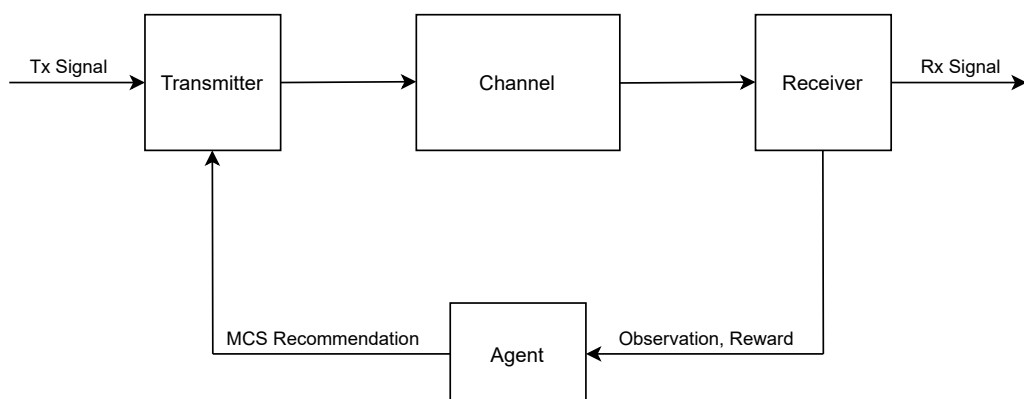

**Figure 2.** The basic concept of the presented approach. An agent observes the state and recommends an action. Finally, the environment provides a reward, depending on the success of the previous transmission.

The agent implements an actor-critic (AC) RL algorithm, while GNU Radio encapsulates the RL environment, as displayed in Figure 3. The environment provides an observable state in form of estimated *SINR* values corresponding to the *n* equidistant data carrying subcarriers of the previous transmission (i.e., without the guardbands and DC carrier).

$$s = \begin{bmatrix} SINR_{SC_0} \\ SINR_{SC_1} \\ \vdots \\ SINR_{SC_n} \end{bmatrix} \tag{1}$$

The action space consists of the available *MCSs* defined by the given radio transmission standard. For this work, our actions space consists of eight actions, as defined by the IEEE 802.11p standard [12].

$$a \in \{MCS0, MCS1, \dots, MCS7\} \tag{2}$$

The packet success rate $PSR = (1 - PER)$ with *PER* indicating the packet error rate, is then used to calculate the effective data rate $\mathcal{R}_{effective}$ that is provided as reward *R* to the agent, where $\mathcal{R}_{a,theoretical}$ is defined as the coded data rate of the chosen *MCS*.

$$\mathcal{R}_{effective} = (1 - PER) \cdot \mathcal{R}_{a,theoretical} \tag{3}$$

The *PSR* can be used for the reward calculation during the training process. Since we do not have access to the receiver's *PER* in a real-world scenario, we propose using a single metric or a combination of several metrics that are available on the transmitter once the training process has terminated successfully, e.g., cyclic redundancy check (CRC).

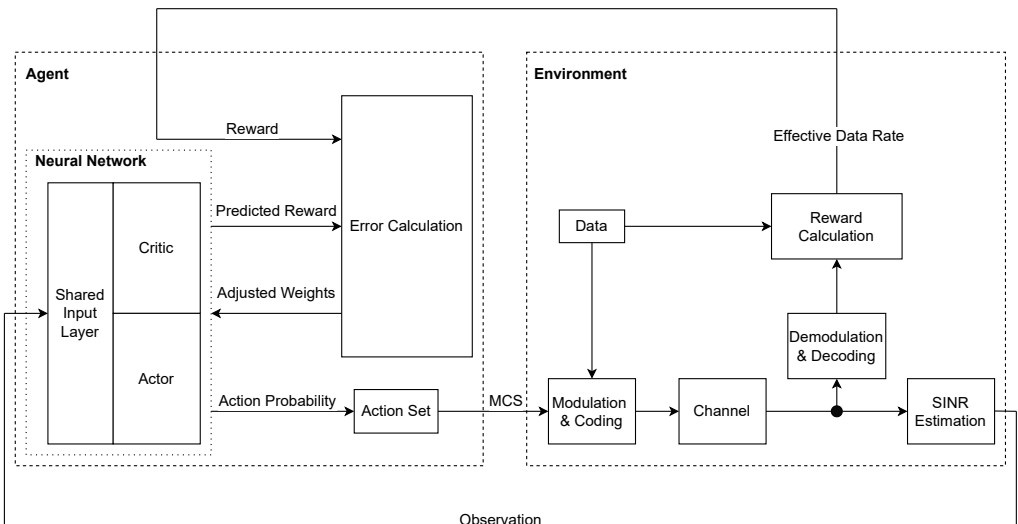

**Figure 3.** Block diagram showing the proposed approach during training with the underlying mechanics of agent and environment.

### 4.1. RL Algorithms

This work considers two different AC algorithms to create an LA approach for dynamic environments.

#### 4.1.1. Natural Actor-Critic (NAC)

Based on the proposed algorithm in [13], the neural network (NN) only consists of three layers: an input layer, a hidden layer, and an output layer. Both the actor and the critic network share a common input layer of size 52, which is the size of our observation space since the IEEE 802.11p standard uses 52 active subcarriers. A hidden layer of size 128 follows. The output layer for the actor consists of eight neurons representing the eight possible actions defined in the action space. A softmax activation function is then applied to the output neurons to model the probability density function of the action space. The agent then samples an action from this distribution to determine the proposed action to apply to the following communication. The critic, on the other hand, has a single output neuron that represents the expected return. All layers are fully connected and use ReLU as an activation function.

#### 4.1.2. Proximal Policy Optimization (PPO)

The AC approach using a PPO algorithm relies on an input layer of size 52, followed by two hidden layers of size 64. As with the NAC approach, PPO uses eight output neurons for the actor network and one output neuron for the critic network. In contrast with the NAC approach, actor and critic networks have different learning rates and use a clipping function. This increases the algorithm's complexity but should improve the performance. Hyperparameters are derived from Schulman et al. [14,15].

### 4.2. Channels

The performance is assessed across three different types of channels to evaluate the effectiveness of the proposed approaches.

#### 4.2.1. Additive White Gaussian Noise (AWGN)

An AWGN channel represents the simplest channel impairment for the experiments presented in this paper. In GNU Radio, a noise vector is sampled from a Gaussian distribution and applied to the transmitted signal. Due to its simplicity, the AWGN channel is used as a baseline for almost optimal performance in the absence of channel impairments.

The performance of the models on the more complex NLOS and real-world channels can then be compared to this baseline to better understand the impact of the different models.

### 4.2.2. Non-Line-of-Sight (NLOS)

The NLOS channel builds upon the AWGN channel. In addition to the added noise on the receiver side, Rayleigh fading based on Clarke's model is added [16]. Additionally, a 3GPP compliant V2X-based attenuation is considered, using an arbitrary distance of 10 m between transmitter and receiver, describing the path loss $PL$ in Equation (4), using the distance in the three-dimensional space $d_{3D}$ and a carrier frequency $f_c$ of 5.9 GHz [17].

$$PL = 36.85 + 30 \log_{10}(d_{3D}) + 18.9 \log_{10}(f_c) \tag{4}$$

### 4.2.3. Real World

A prerecorded channel is used to test the performance of the approach using real-world channel impairments [18,19]. A massive multiple-input and multiple-output (mMIMO) antenna array (64 elements) was installed on a building's roof approximately 20 m above the ground. A trolley carrying the receive antenna was pushed along the ground to imitate a pedestrian. A 2.180 GHz carrier frequency was used with 50 sub-bands over a bandwidth of 10 MHz for each antenna, and measurement impairments were eliminated by post-processing. To support our experiments using the IEEE 802.11p standard, we linearly interpolate the channel in the time and frequency domains to simulate the receive antenna moving between 60 km/h and 100 km/h. The prerecorded real-world channel is used to provide insights into how performant our approach is on non-simulated channels. As the intention is to use this approach to improve communication over real-world channels, it is important to include other possible impediments not captured by more simplistic channel models. For our experiments, we consider a pair from the MIMO channel recordings for our channel, as we are simulating a single-input and single-output (SISO) environment.

## 5. Results

We performed several experiments with different channel impairments and algorithms to evaluate both AC RL approaches. Table 1 provides the theoretical data rates as a point of comparison for our experiments. The evaluation focuses on analyzing the impact of the channel and chosen RL algorithm.

**Table 1.** MCS values with associated coded data rate (CDR), modulation, code rate (CR), and coded bit rate (CBR) for the IEEE 802.11p OFDM PHY layer at a bandwidth of 10 MHz, which represents the used action space for each of the following experiments [20].

| MCS | CDR in Mbps | Modulation | CR | CBR in Mbps |
|-----|-------------|------------|-----|-------------|
| 0 | 3 | BPSK | 1/2 | 6 |
| 1 | 4.5 | BPSK | 3/4 | 6 |
| 2 | 6 | QPSK | 1/2 | 12 |
| 3 | 9 | QPSK | 3/4 | 12 |
| 4 | 12 | 16QAM | 1/2 | 24 |
| 5 | 18 | 16QAM | 3/4 | 24 |
| 6 | 24 | 64QAM | 2/3 | 36 |
| 7 | 27 | 64QAM | 3/4 | 36 |

The algorithm is trained from scratch each time using a uniform weight initialization and applied on an IEEE 802.11p flowgraph with a bandwidth of 10 MHz. Based on the observed state $S$ of the previous transmission, the agent produces a probability distribution over the entire action space, which will be referred to as *the recommendation*. After sampling from the recommendation, the chosen MCS is applied to a simulated transmission in the GNU Radio environment. The agent then receives the reward $R$ as described in Section 4.

The reward graph displays the effective data rate in Mbps and is used as the main metric for the comparisons between the different approaches. The effective data rate is calculated as the PSR multiplied with the encoded bit rate. Therefore, only correctable packets and detected packets are considered, while non-correctable packets and non-detected ones count as a loss. In addition, the action selection process is shown on the right-hand side of each plot. It covers the absolute number of steps per episode in which an action *A* was chosen. At the beginning of each experiment, the recommended actions are approximately equally distributed within an episode, indicating the agent is guessing and cannot rely on any experience. After a few episodes, the agent begins to recognize more promising MCSs and adapts its recommendations accordingly until the count of some actions increase while others decrease noticeably. This metric provides further insights on how the effective data rate is achieved.

The training procedure iterates for 500 training episodes with 50 steps each, during which the agent's performance typically converges. After each episode, an episodic reward can be determined, which contributes to the smoothed running reward.

### 5.1. AWGN Channel

Figure 4 displays the results of the NAC learning process on an AWGN channel at a static SNR level of 15 dB. The agent converged towards choosing action 5 (MCS5), resulting in the reward of approximately 15.3 Mbps. For comparison, the theoretical data rate for MCS4 and MCS5 lies at 12.0 Mbps and 18 Mbps, respectively.

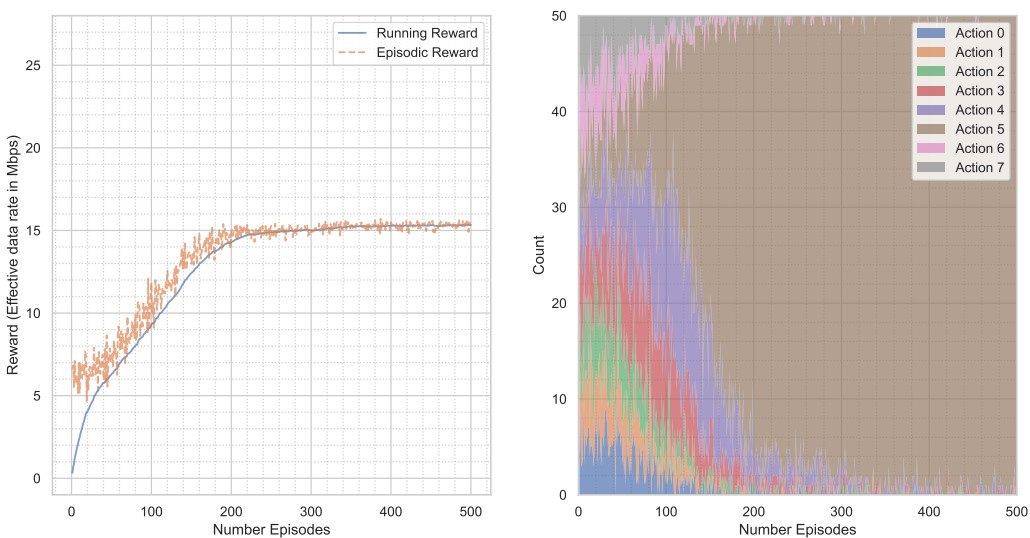

**Figure 4.** The behavior of the NAC reinforcement learning approach on an AWGN channel at a static SNR level of 15 dB.

A second experiment was performed, in which the SNR of the AWGN channel increased by 1 dB after each step. At the end of each episode, the SNR was set to zero. After 500 episodes, the agent achieved an effective data rate of approximately 16 Mbps. Figure 5 shows the proposed episodical combination of MCS5, MCS6, and MCS7 with theoretical limits of 18 Mbps, 24 Mbps, and 27 Mbps, respectively. This result hints that the algorithm recognizes the benefits of applying a lower MCS index to lower SINR observations, while for sufficient channel conditions, the highest MCS index can be used to provide an increased data rate.

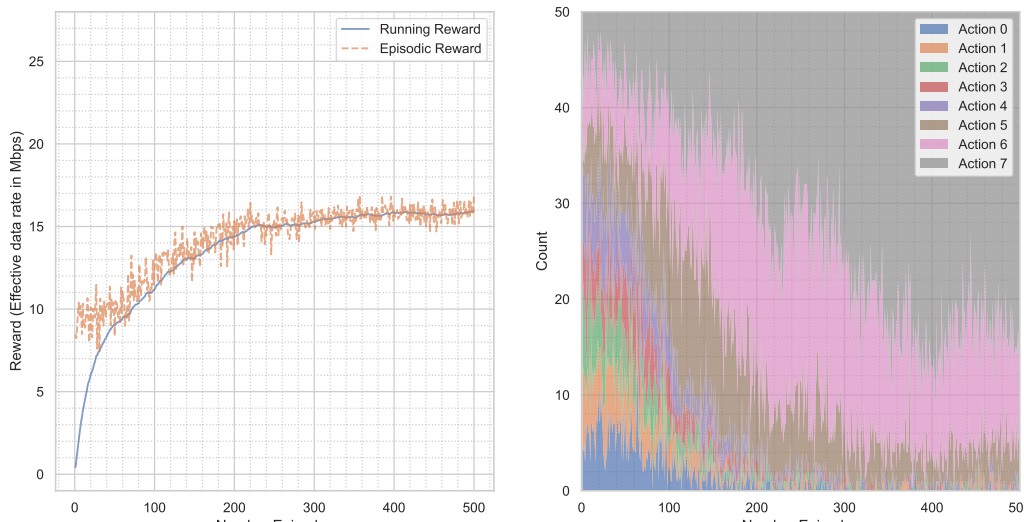

**Figure 5.** The behavior of the NAC reinforcement learning approach on an AWGN channel with a sweeping SNR level between 0 dB and 50 dB.

### 5.2. NLOS Channel

Figure 6 shows the result of using the NAC approach on an NLOS channel with a static SNR level of 15 dB, fading channel effects, and attenuation caused by a 10 m path loss simulation. The agent chose a combination of MCS5, MCS6, and MCS7 with theoretical data rates of 18 Mbps, 24 Mbps, and 27 Mbps, respectively. The achieved effective data rate is approximately 18.75 Mbps. A combination of MCSs is uncommon for typical LA implementations on almost non-changing channel states. However, in this experiment, the agent estimates that a more granular differentiation is beneficial to achieve a high effective data rate.

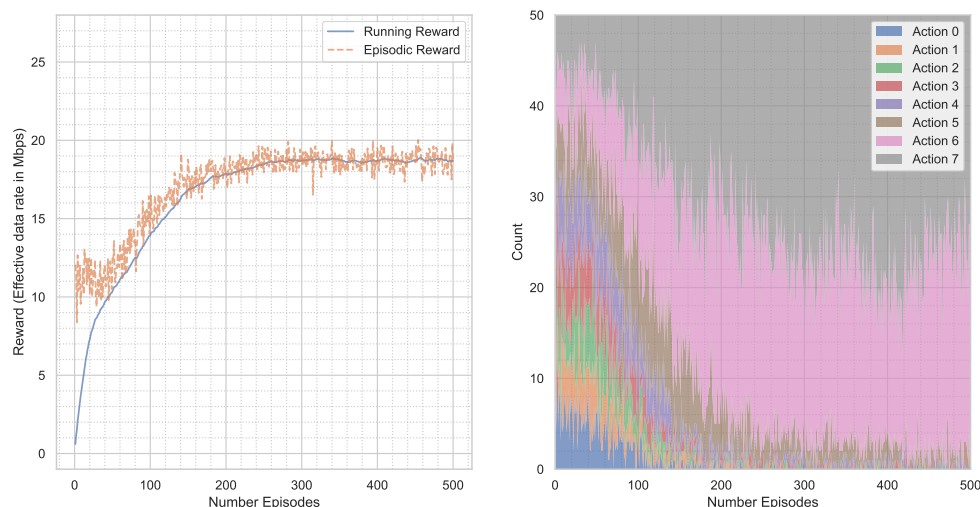

**Figure 6.** The behavior of the NAC reinforcement learning approach on a Rayleigh NLOS channel at a static SNR of 15 dB.

The experiments on the NLOS channel show similar results to the outcome produced on the AWGN channel, despite the added Rayleigh fading and attenuation. By looking at the reward, results in Figure 6 show an effective data rate of approximately 18.75 Mbps and outperform results in Figure 4, which cover an effective data rate of approximately 15.3 Mbps, as the learned model proposes a combination of several MCSs per episode rather than recommending a single one. This is likely thanks to a decrease in the learning rate in this experiment from $10^{-2}$ to $10^{-3}$.

### 5.3. Real-World Channel

The real-world channel presented in [18,19] includes several types of impairments, i.e., the prerecorded channel effects, attenuation, and AWGN on the receiver side. Figure 7 displays the outcome produced on the real-world channel with its prerecorded channel effects, a static AWGN level of 15 dB, and an attenuation caused by a 10 m path loss. There is a noticeable spike in episodic reward right at the beginning of the recording. This effect occurs in every rerun of the experiment and is potentially caused by a gap in the data recording that includes reduced impediments. The agent mostly chose MCS7 for its recommendation, with a coded data rate of 27 Mbps. According to the curve that displays the effective data rate, a throughput of almost 16 Mbps was achieved, which is far below the theoretical data rate of 27 Mbps. With an increase in the static SNR level to 50 dB, as shown in Figure 8, the effective data rate increases notably to more than 26 Mbps. This hints that the newly introduced channel effects impair the achievable throughput but do not impact the agent's decision-making. Increasing the SNR decreases the impact of the prerecorded channel effects, and a data rate close to the optimum can be reached.

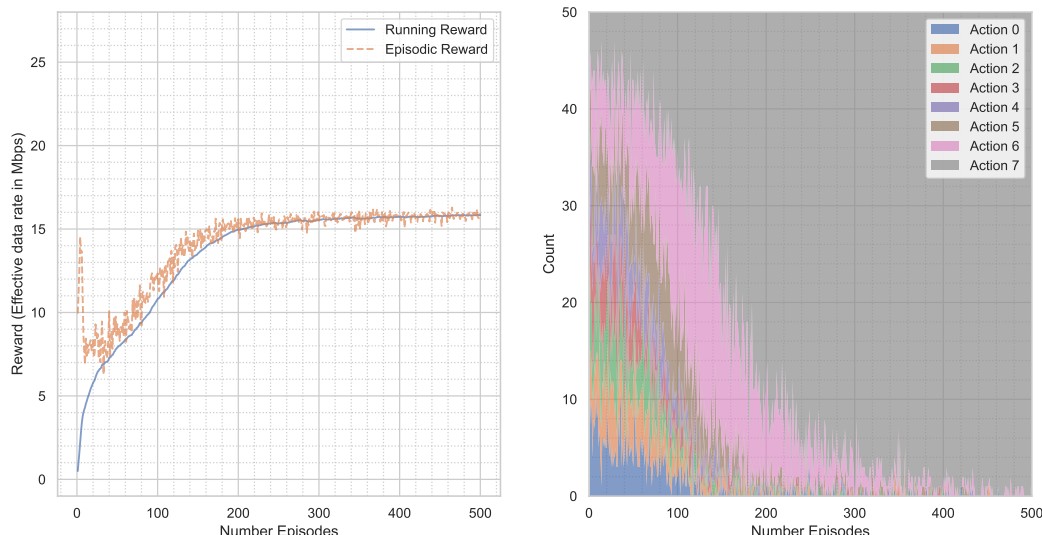

**Figure 7.** The behavior of the NAC reinforcement learning approach on a real-world channel at a static SNR level of 15 dB and a spike in episodic reward right at the beginning.

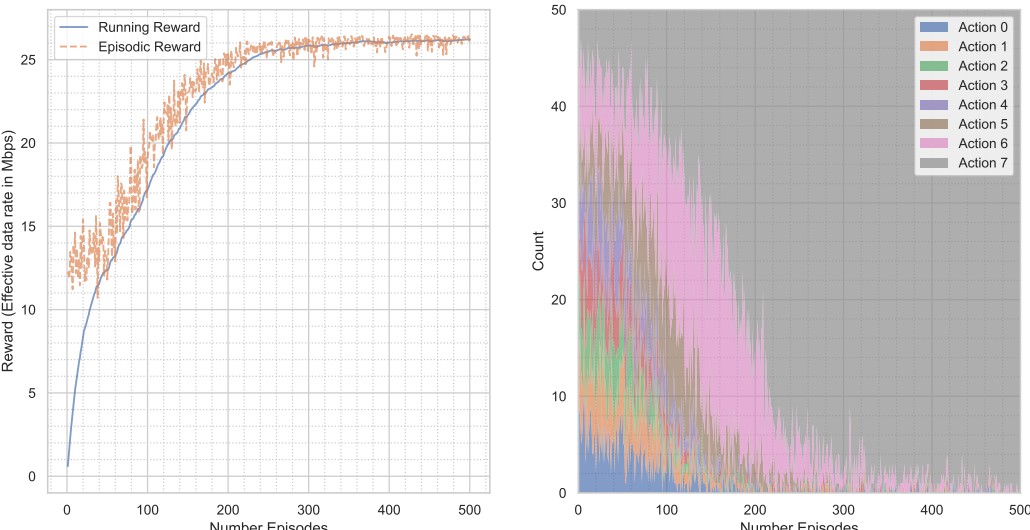

**Figure 8.** The behavior of the NAC reinforcement learning approach on a real-world channel at a static SNR level of 50 dB.

### 5.4. PPO-Algorithm

To evaluate the impact of the choice of the AC method, we also evaluated the PPO algorithm. The AWGN channel was chosen to facilitate the common ground for comparison due to its focus on a single channel impairment. Figure 9 displays the behavior of a PPO approach on an AWGN channel with a static SNR of 15 dB. The agent almost exclusively chooses action 5 (MCS5), resulting in a reward of 15.5 Mbps compared to the theoretical data rate of 18 Mbps.

The PPO algorithm results in a convergence in approximately 160 episodes, down from the 300 episodes of the NAC approach on the same channel. Furthermore, the variance of the episodic reward diminishes, and the algorithm converges to a single MCS recommendation faster. In contrast with the NAC RL algorithm on the 15 dB AWGN channel, the agent chooses to stick with MCS5 instead of MCS7.

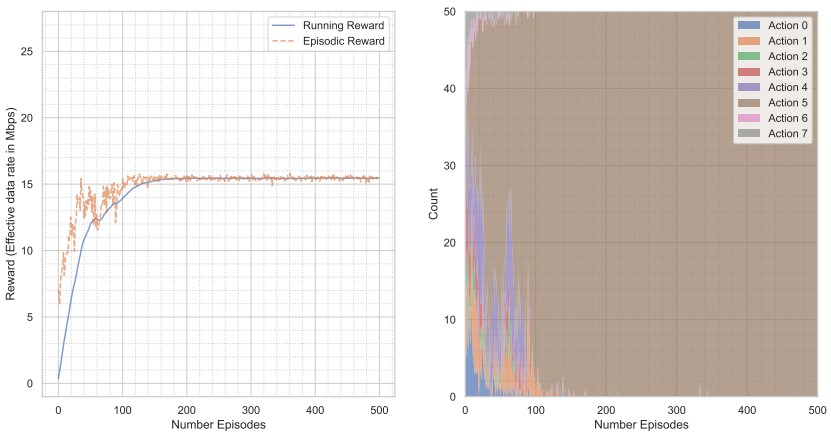

**Figure 9.** The behavior of the PPO reinforcement learning approach on an AWGN channel with a static SNR of 15 dB.

## 6. Discussion

The presented DRLLA approach does not directly optimize for low PER but for data rate, while the PER adjusts according to the channel state. IEEE 802.11p, on the other hand, is optimized for reliability with PER as a fixed parameter, while the throughput then adjusts according to the channel quality. Therefore, due to these different objectives, neither the throughput nor reliability of the produced results can be directly compared to the state of the art.

The results of the experiments carried out in Section 5 exhibit that DRLLA can be applied to the various examined channels. When a clear decision is possible, the agent recommends almost solely a single MCS during the entire episode of a static SNR level experiment, as expected and performed by traditional LA algorithms. Whenever the system reaches an SNR level where the transmission could benefit from a combination of more than one MCS, the agent suggests up to three primary actions to be used within the same episode, which is a noticeable difference from typical LA algorithms. As the sweeping SNR experiments hint, DRLLA is able to quickly adapt to changing channel states.

DRLLA approaches the theoretical data rate on the AWGN and NLOS channels well. Even on the real-world channel, DRLLA manages to establish a transmission rate of almost 16 Mbps at a static SNR level of 15dB. Increasing the SNR to 50 dB improves the data rate heavily, hinting that some channel effects included in the prerecorded channel are not covered by the observation and action space but can be tackled with a higher SNR. Stated channel effects are the cause of a slightly higher PER compared to the AWGN and NLOS channel.

The sometimes relatively large gap between the theoretical data rate and the effective data rate of an MCS hint that the agent prefers a high throughput over a low PER. This can be attributed to the chosen reward function. The difference in theoretical data rate between

the individual MCS is rather large in the IEEE 802.11p definition. Therefore, a single packet transmitted with a higher MCS index is often valued higher than several successfully transmitted packets with lower MCS indexes. Since our reward function focuses on high throughput, the agent's propensity towards higher MCS indexes despite potentially not matching the channel quality makes sense.

In contrast to the state of the art for LA, the proposed method benefits from multiple factors, including the experience or knowledge stored in its neural networks, its capability to learn the characteristics of the environment, and the ability to adapt to unseen conditions over time.

Even though DRLLA adds some computational complexity, the advantages over traditional techniques outweigh the drawbacks of additional overhead, especially since modern hardware and model compression techniques can greatly reduce the runtime.

## 7. Conclusions and Future Work

In this paper we examined a deep reinforcement learning method for link adaptation and tested several algorithms on different channels. In order to properly leverage the promising advantages of DRLLA, we propose the following directions for future improvements. First, the current reward function optimizes for throughput. However, as we saw in our experiments, this can lead to other transmission metrics taking a hit. When applying LA in real-world scenarios, some use cases can benefit from a balance of different transmission metrics, including latency, PER, and data rate. For a more balanced implementation, we can derive a more complex reward function that considers additional characteristics of the transmission system. The achieved data rate may drop slightly, but the overall system can potentially benefit from other application-specific objectives.

Additionally, a more sophisticated sampling method can be used to interpret the produced probability distribution (e.g., only sample from the actions with the top k probabilities). The field of generative language modeling proposes many different approaches, such as Top-K [21] or Top-P [22] sampling methods. This would prevent the extremely low likelihood, suboptimal actions from being used due to chance.

Also, the observation space can be increased to further enhance the agent's decision-making process. Observing additional channel-related metrics might improve DRLLA's performance due to the agent's increased knowledge about the state of the system.

Finally, to close the gap with the state-of-the-art mobile telecommunication networks and to gain from underlying mechanics that improve a real-world outdoor radio transmission besides LA, an upgrade of the radio simulation flowgraph from IEEE 802.11p to 4G LTE or 5G NR can be beneficial. This enhancement potentially increases the DRLLA performance on the real-world channel without touching the LA part.

In conclusion, this paper presents the potential of a DRL-based approach for LA called DRLLA and identifies potential advantages when compared to the state of the art. Results hint that the reward function plays a major role in this work and could be used to adjust for one or several application-related objectives. Thus, this can be a potential detail to continue future research.

**Author Contributions:** Investigation, F.G.; Methodology, F.G. and D.W. (Daniel Wessel); Software, F.G.; Supervision, D.W. (Daniel Wessel) and A.V.; Validation, A.V.; Writing—original draft, F.G.; Writing—review and editing, D.W. (Daniel Wessel), M.H., A.W., D.W. (Dirk Wübben), A.D. and A.V. All authors have read and agreed to the published version of the manuscript.

**Funding:** This research was partly funded by German Ministry of Education and Research (BMBF) under grant 16KIS1184 (FunKI).

**Data Availability Statement:** Not applicable.

**Conflicts of Interest:** The authors declare no conflict of interest. The funders had no role in the design of the study; in the collection, analyses, or interpretation of data; in the writing of the manuscript, or in the decision to publish the results.

**Abbreviations**

The following abbreviations are used in this manuscript:

| | |
|---|---|
| AC | Actor Critic. |
| AWGN | Additive White Gaussian Noise. |
| CBR | Coded Bit Rate. |
| CCT | CQI Correction Term. |
| CDR | Coded Data Rate. |
| CQI | Channel Quality Indicator. |
| CR | Code Rate. |
| CRC | Cyclic Redundancy Check. |
| DRL | Deep Reinforcement Learning. |
| DRLLA | Deep Reinforcement Learning for Link Adaptation. |
| GR | GNU Radio. |
| HMM | Hidden Markow Model. |
| LA | Link Adaptation. |
| LTS | Latent Thompson Sampling. |
| MCS | Modulation and Coding Scheme. |
| MIMO | Multiple Input and Multiple Output. |
| mMIMO | Massive MIMO. |
| ML | Machine Learning. |
| NAC | Natural Actor Critic. |
| NLOS | Non-Line of Sight. |
| NN | Neural Network. |
| NR | New Radio. |
| OFDM | Orthogonal Frequency-Division Multiplexing. |
| OLLA | Outer Loop Link Adaptation. |
| PDF | Probability Density Function. |
| PER | Packet Error Rate. |
| PPO | Proximal Policy Optimization. |
| PSR | Packet Success Rate. |
| RL | Reinforcement Learning. |
| RSSI | Received Signal Strength Indicator. |
| SC | Sub carrier. |
| SINR | Signal to Interference plus Noise Ratio. |
| SISO | Single Input and Single Output. |
| SNR | Signal to Noise Ratio. |
| TROLL | Training of Outer Loop Link Adaptation. |
| UE | User Equipment. |

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
