# Peer review of "DRLLA: Deep Reinforcement Learning for Link Adaptation"

_telecom, doi:10.3390/telecom3040037_

Round 1

Reviewer 1 Report

This work is interesting. Could it be applied for satellite communication also? I consider the satellite link more important nowadays because the link is changing faster. 

The authors mentioned "traditional LA algorithms" could they include a comparative table with the traditional and new algorithms used?

An acronymous list is recommended to include.

Author Response

Review: This work is interesting. Could it be applied for satellite communication also? I consider the satellite link more important nowadays because the link is changing faster.
Response: The work can be applied to any Link Adaptation problem, as long as valuable observations can be provided to the agent and there is an action space from which the algorithm can choose.

Review: The authors mentioned "traditional LA algorithms" could they include a comparative table with the traditional and new algorithms used?
Response:  Unfortunately, the traditional algorithm can not directly be compared to the DRLLA approach in a fair way, due to the different objectives (as mentioned in the Discussion section). While the PER is a fixed parameter in the standard approach for IEEE 802.11p, the presented DRLLA method optimizes for throughput and neglects the PER.

Review: An acronymous list is recommended to include.
Response: A list of abbreviations has been added

Reviewer 2 Report

Results and discussing the results should be separated from the conclusion.  I suggest that conclusion and future work should be combined and to put the results along with discussion to make it more constructive.

Author Response

Review: Results and discussing the results should be separated from the conclusion.  I suggest that conclusion and future work should be combined and to put the results along with discussion to make it more constructive.
Response: Conclusion and future work have been combined while discussion got its own section.

Reviewer 3 Report

Thank you for your well-written paper. I enjoyed reading about this application of reinforcement learning in the field; I believe the document is well structured, written, and of adequate length. 

I only have minor corrections and comments for the authors. 

I would have liked to see more in-depth discussions regarding why you decided to use the NAC and PPO RL algorithms in your experiments and  a quantitative performance comparison between the two simulated outcomes, perhaps in a summary table. The authors mention running the agents in the field; what would be an estimate of the minimum computational resources needed to run it at the "edge"? Would network quantization be required? 

Figures 5 - 9: The font size of the axes' labels is too small when printed out. Also, please increase the line width of the reward evolution. (Minor change, feel free to skip if too tricky to make). 

Lines 142 & 354 Please replace or delete '/' with "and" or "or" to reduce vagueness. 

Thank you. 

Author Response

Review: Thank you for your well-written paper. I enjoyed reading about this application of reinforcement learning in the field; I believe the document is well structured, written, and of adequate length.
I only have minor corrections and comments for the authors.
Response: Thank you very much!

Review: I would have liked to see more in-depth discussions regarding why you decided to use the NAC and PPO RL algorithms in your experiments and a quantitative performance comparison between the two simulated outcomes, perhaps in a summary table. The authors mention running the agents in the field; what would be an estimate of the minimum computational resources needed to run it at the "edge"? Would network quantization be required?
Response: Minimum computational resources for DRL are hard to tell and depend on various things, such as whether inference or training is happening on the edge. Instead of a very vague guess, the authors decided to not include speculations and leave this point open for future work. Network quantization is not necessarily needed but could lower the hardware requirements. 

Review: Figures 5 - 9: The font size of the axes' labels is too small when printed out. Also, please increase the line width of the reward evolution. (Minor change, feel free to skip if too tricky to make).
Response: Skipped due to effort & deadline (has already been postponed once)

Review: Lines 142 & 354 Please replace or delete '/' with "and" or "or" to reduce vagueness.
Response: '/' have been replaced